# Anti-Glycolipid Antibody Examination in Five EAE Models and Theiler’s Virus Model of Multiple Sclerosis: Detection of Anti-GM1, GM3, GM4, and Sulfatide Antibodies in Relapsing-Remitting EAE

**DOI:** 10.3390/ijms241612937

**Published:** 2023-08-18

**Authors:** Kota Moriguchi, Yumina Nakamura, Ah-Mee Park, Fumitaka Sato, Motoi Kuwahara, Sundar Khadka, Seiichi Omura, Ijaz Ahmad, Susumu Kusunoki, Ikuo Tsunoda

**Affiliations:** 1Department of Microbiology, Faculty of Medicine, Kindai University, Osakasayama City 589-8511, Osaka, Japan; moriguchi.ndmc@gmail.com (K.M.); 76.yumipia.sakura0226@gmail.com (Y.N.); ampk@med.kindai.ac.jp (A.-M.P.); fsato@med.kindai.ac.jp (F.S.); cls.sundar@gmail.com (S.K.); somura@med.kindai.ac.jp (S.O.); ijazahmad383@gmail.com (I.A.); 2Department of Internal Medicine, Japan Self Defense Forces Hanshin Hospital, Kawanishi City 666-0024, Hyogo, Japan; 3Department of Life Science, Faculty of Science and Engineering, Kindai University, Higashiosaka City 577-8502, Osaka, Japan; 4Department of Arts and Science, Faculty of Medicine, Kindai University, Osakasayama City 589-8511, Osaka, Japan; 5Department of Neurology, Faculty of Medicine, Kindai University, Osakasayama City 589-8511, Osaka, Japan; kuwahara@med.kindai.ac.jp (M.K.); kusunoki-susumu@jcho.go.jp (S.K.); 6Department of Immunology, School of Medicine, Duke University, Durham, NC 27710, USA; 7Japan Community Health care Organization (JCHO) Headquarters, Minato City 108-8583, Tokyo, Japan

**Keywords:** animal models, autoimmunity, CNS demyelinating diseases, cytokines, enzyme-linked immunosorbent assay, neuroimmunology, neuroinflammatory diseases, neurovirology, *Picornaviridae* infections

## Abstract

Anti-glycolipid antibodies have been reported to play pathogenic roles in peripheral inflammatory neuropathies, such as Guillain–Barré syndrome. On the other hand, the role in multiple sclerosis (MS), inflammatory demyelinating disease in the central nervous system (CNS), is largely unknown, although the presence of anti-glycolipid antibodies was reported to differ among MS patients with relapsing-remitting (RR), primary progressive (PP), and secondary progressive (SP) disease courses. We investigated whether the induction of anti-glycolipid antibodies could differ among experimental MS models with distinct clinical courses, depending on induction methods. Using three mouse strains, SJL/J, C57BL/6, and A.SW mice, we induced five distinct experimental autoimmune encephalomyelitis (EAE) models with myelin oligodendrocyte glycoprotein (MOG)_35–55_, MOG_92–106_, or myelin proteolipid protein (PLP)_139–151_, with or without an additional adjuvant curdlan injection. We also induced a viral model of MS, using Theiler’s murine encephalomyelitis virus (TMEV). Each MS model had an RR, SP, PP, hyperacute, or chronic clinical course. Using the sera from the MS models, we quantified antibodies against 11 glycolipids: GM1, GM2, GM3, GM4, GD3, galactocerebroside, GD1a, GD1b, GT1b, GQ1b, and sulfatide. Among the MS models, we detected significant increases in four anti-glycolipid antibodies, GM1, GM3, GM4, and sulfatide, in PLP_139–151_-induced EAE with an RR disease course. We also tested cellular immune responses to the glycolipids and found CD1d-independent lymphoproliferative responses only to sulfatide with decreased interleukin (IL)-10 production. Although these results implied that anti-glycolipid antibodies might play a role in remissions or relapses in RR-EAE, their functional roles need to be determined by mechanistic experiments, such as injections of monoclonal anti-glycolipid antibodies.

## 1. Introduction

Glycolipids are components of the cell membrane and encompass a wide variety of compounds, including glycosphingolipids such as cerebrosides, gangliosides, and sulfatides [1]. Glycosphingolipids have a glycan structure attached to a lipid tail that contains ceramide; particularly, gangliosides are glycosphingolipids containing one or more sialic acid residues [2] and are highly abundant glycolipids in the nervous system [3]. Although the heterogeneity in the sugar compositions of the glycan headgroup can give more than 200 ganglioside structures theoretically [2], the bulk of glycolipids in vertebrates is composed of only a few major glycolipid classes. Structures of representative glycolipids are shown in Figure 1 [4]. These glycolipids are included in both the peripheral nervous system and the central nervous system (CNS) with different compositions [5,6,7]. Antibodies to glycolipids have been detected in sera from immune-mediated peripheral nerve diseases, including Guillain–Barré syndrome (GBS) and Fisher syndrome [8]. Several individual anti-glycolipid antibodies can be useful diagnostic markers and have been suggested to play pathogenic roles, since these antibodies were often associated with specific clinical signs/symptoms.

Multiple sclerosis (MS) is a chronic inflammatory demyelinating disease in the CNS, where autoimmune responses to the CNS components have been proposed to damage the CNS tissues, resulting in demyelination and axonal degeneration [9]. MS has been classically divided into several subtypes by the clinical courses [10], including relapsing-remitting MS (RR-MS), primary progressive MS (PP-MS), and secondary progressive MS (SP-MS) [11,12]. RR-MS is defined as episodes of neurologic attacks with total or partial recovery. PP-MS is steadily worsening of the neurologic signs following the initial attack. SP-MS is steadily worsening of the neurologic signs following the initial RR-MS course. Although the prognosis and treatment of MS subtypes differ, there are few biomarkers or pathomechanisms that can distinguish or explain the differences among the MS subtypes. Although anti-ganglioside antibodies in MS were first described in 1980, it has been unclear whether anti-glycolipid antibodies could be used as biomarkers or whether anti-glycolipid antibodies play a pathogenic or protective role in MS [13]. In MS patients, anti-ganglioside antibodies were reported to be frequently elevated (3–48%), although the sensitivity and specificity were low, compared with autoimmune neuropathies [14]. On the other hand, Sadatipour et al. [15] reported that anti-ganglioside antibodies were associated with MS subtypes; the percentages of plasma samples with increased anti-GM3 antibody were significantly higher in patients with PP-MS (56.3%) and SP-MS (42.9%) than in those with RR-MS (2.9%) and healthy controls (2.6%).

Although the precise etiology of MS is unclear, anti-myelin autoimmunity and viral infections have been proposed to trigger MS. The autoimmune and viral etiologies have been supported by the two commonly used animal models of MS: experimental autoimmune encephalomyelitis (EAE) and Theiler’s murine encephalomyelitis virus (TMEV)-induced demyelinating disease (TMEV-IDD) [16]. EAE is an autoimmune model of MS and can be actively induced in animals by subcutaneous sensitization with myelin antigens in complete Freund’s adjuvant (CFA), with or without an additional adjuvant. Various clinical courses of human MS can be reproducible in several EAE models by changing the experimental conditions, such as animal species, antigens, and adjuvants [17] (Table 1). For example, in mice, the RR disease course can be induced in SJL/J mice by sensitizing with myelin proteolipid protein (PLP) and myelin oligodendrocyte glycoprotein (MOG) or their encephalitogenic peptides, including the PLP_139–151_ peptide [18]. Co-injection of the additional adjuvant curdlan with the PLP_139–151_ peptide in SJL/J mice resulted in the induction of hyperacute fatal EAE. On the other hand, the PP and SP disease courses can be induced by co-injection of curdlan in MOG_92–106_-sensitized SJL/J mice [19], although MOG_92–106_ sensitization alone induced PP-EAE in A.SW mice. MOG_92–106_-sensitized A.SW mice developed ataxia (“ataxic EAE”) instead of motor paralysis [20], although mice in other EAE models usually developed “classical EAE”, characterized by ascending motor paralysis. The most widely used EAE model is MOG_35–55_-sensitized C57BL/6 mice with additional injections of pertussis toxin (PT); the mice developed monophasic EAE with no relapse, which may correspond to acute disseminated encephalomyelitis (ADEM) or clinically isolated syndrome (CIS) in humans. The remission and severities of MOG_35–55_-induced C57BL/6 mice have been shown to differ depending on the experimental conditions, such as the number of MOG_35–55_-sensitization [21]. Although the standard MOG_35–55_-sensitization resulted in mild EAE with complete remission, more aggressive EAE induction led to more severe EAE with incomplete recovery during the chronic stage (“chronic EAE”). Myelin-specific CD4^+^ T cells play an effector role in most EAE models; anti-myelin antibodies have also been shown to play a pathogenic role in some EAE models [22].

MS-like diseases can also be induced by inoculation of viruses in animals. TMEV-IDD is the most widely used viral model for MS. TMEV is a single-stranded (+) RNA virus that belongs to the family *Picornaviridae* [24]. Intracerebral inoculation of TMEV resulted in a biphasic disease: during the acute phase, 1 week post infection, TMEV-infected mice developed acute polioencephalitis, but recovered completely in 2 weeks. Around 1 month after TMEV infection, the mice developed TMEV-IDD, a primary progressive inflammatory demyelinating disease with viral persistence, whose pathology was similar to MS [21]. In TMEV-IDD, both humoral and cellular immune responses have been shown to play pathogenic roles for demyelination [25]. Although most mouse strains were resistant to TMEV infection, only a few susceptible strains, including SJL/J, developed TMEV-IDD.

The relevance between MS and anti-glycolipid antibodies has rarely been reported; the etiological significance of anti-glycolipid antibodies still needs to be determined. A comprehensive study to research anti-glycolipid antibodies in animal models has yet to be performed. Thus, we examined whether anti-glycolipid antibodies could be detected in murine MS models with different disease courses (RR, SP, PP, and chronic) and etiologies (autoimmune versus viral). We found that only PLP_139–151_-sensitized mice with RR-EAE had serum anti-GM1, GM3, and GM4 antibodies, although the role of the antibodies in remissions or relapses was unclear.

## 2. Results

### 2.1. Mouse Models of MS with Distinct Clinical Courses

We induced five EAE models using three different myelin antigen peptides and three mouse strains (Figure 2). PLP_139–151_-sensitised SJL/J mice developed RR-EAE with an initial attack around 10 days post-induction (p.i.) and multiple relapses and remissions during the 2-month observation period. With curdlan co-injection, PLP_139–151_-sensitized SJL/J mice developed hyperacute EAE and died around 2 weeks p.i. [the mean survival days ± standard error of the mean (SEM) of hyperacute EAE mice was 15.1 ± 2.3], although MOG_92–106_-sensitised SJL/J mice developed PP-EAE or SP-EAE in 1–2 months p.i. On the other hand, without curdlan injection, MOG_92–106_-sensitised A.SW mice developed PP-EAE with a late disease onset; all mice died of continuous disease progression with no remission. By the aggressive EAE induction method, MOG_35–55_-sensitised C57BL/6 developed chronic EAE 1 month p.i. TMEV-infected mice developed a chronic demyelinating disease, TMEV-IDD, around 1 month p.i., whose disease course was primary progressive (PP).

### 2.2. Anti-Glycolipid Antibodies in PLP_139–151_-Sensitized SJL/J Mice with RR-EAE

When we completed monitoring MS model mice, we harvested sera from SJL/J mice sensitized with PLP_139–151_ or MOG_92–106_ (with curdlan co-injection), or infected with TMEV; MOG_35–55_-sensitized C57BL/6 mice; and MOG_92–106_-sensitized A.SW mice. We also collected sera from untreated SJL/J, C57BL/6, and A.SW mice as controls. Using anti-mouse F(ab’)_2_ antibody that detects all immunoglobulin (Ig) subclasses, we conducted enzyme-linked immunosorbent assays (ELISAs) to determine anti-glycolipid antibody levels. In Figure 3, we compared the levels of antibodies against 10 glycolipids: GM1, GM2, GM3, GM4, GD3, galactocerebroside (GC), GD1a, GD1b, GT1b, and GQ1b among the controls and MS model mice. We found significantly higher levels of anti-GM1, anti-GM3, and anti-GM4 antibodies only in PLP_139–151_-EAE mice with an RR disease course in SJL/J mice, but not in C57BL/6 or A.SW mice (Figure 3). All three control mouse strains did not have substantial anti-glycolipid antibodies, although the levels of anti-GC antibody were slightly higher than those of other glycolipid antibodies (Figure 3). On the other hand, anti-sulfatide antibody levels were high in all control and MS model mice (Figure 4A).

Among the MS models that we used in this study, the pathophysiologies of the PP-EAE model induced by MOG_92–106_ in A.SW mice and the TMEV model have been shown to change during the time course [19]. Thus, using serum samples harvested at early time points of these two models, we quantified representative anti-glycolipid antibodies: anti-GM1, anti-GM3, anti-GM4, and anti-GC antibodies. We detected no or low levels of these anti-glycolipid antibodies in these two models [PP-EAE in A.SW mice (days 12 and 21 p.i., mean absorbance ± SEM): anti-GM1, 0.01 ± 0.03; anti-GM3, 0.1 ± 0.04; anti-GM4, 0.1 ± 0.03; and anti-GC, 0.1 ± 0.03. TMEV model (day 7 after infection): anti-GM1, 0 ± 0; anti-GM3, 0 ± 0; anti-GM4, 0 ± 0; and anti-GC, 0.002 ± 0.005]. Since anti-glycolipid antibody responses were observed mostly in PLP_139–151_-induced EAE mice, we focused on this model in the following experiments.

### 2.3. Anti-Glycolipid Antibodies in the Initial Stage of PLP_139–151_-Induced EAE

PLP_139–151_-induced RR-EAE mice had the first neurological signs around 10 days p.i. (disease onset of the initial stage of EAE), peaked around 2 weeks p.i., and recovered completely in a few days (complete remission), and then later, had multiple remissions and relapses during the time course (Figure 2A). We examined whether anti-glycolipid antibodies could be detectable during the early stage of RR-EAE, 14–21 days p.i., where EAE scores ranged from 0 to 3.5. Anti-GM1, anti-GM3, and anti-GM4 antibodies were detectable in most mice (Figure 5A–C). Low or no antibody levels in one asymptomatic mouse (EAE score = 0) could be due to failure in EAE induction; intriguingly, these antibodies were also undetectable or low in EAE mice with high disease severity (EAE score 3 or higher) (Figure 5A–C). Anti-sulfatide antibodies were higher than the other anti-glycolipid antibodies regardless of disease severity (Figure 5D). We conducted time course studies and Ig subclass analyses of anti-glycolipid antibodies, using IgM-specific or IgG-specific anti-mouse detection antibody. We detected the IgM subclass of anti-sulfatide antibodies as early as 14 days p.i. (Figure 4B). The other anti-glycolipid IgM or IgG antibodies were undetectable or very low (mean absorbance ± SEM: anti-GM1 IgM, 0 ± 0; anti-GM3 IgM, 0.015 ± 0.007; anti-GM4 IgM, 0 ± 0; anti-GM1 IgG, 0 ± 0; anti-GM3 IgG, 0.003 ± 0.002; and anti-GM4 IgG, 0 ± 0).

### 2.4. Increased CD1d-Independent Anti-Sulfatide Lymphoproliferative Responses

In the above experiments, we found higher levels of antibody responses to GM1, GM3, GM4, and sulfatide in PLP_139–151_-induced EAE mice than in the control mice. Since glycolipids have been shown to be presented on CD1d molecules [26], we tested whether PLP_139–151_-induced EAE mice could have cellular immune responses to the four glycolipids via CD1d molecules. We also tested the immune response to GD1b, since anti-GD1b immune response has been reported in EAE [27]. We isolated mononuclear cells (MNCs) from PLP_139–151_-induced EAE mice and conducted lymphoproliferative assays using the Cell Counting Kit-8 (CCK-8) (Figure 6). We found that PLP_139–151_-induced EAE mice had significantly higher lymphoproliferative responses to sulfatide, but not to the other four glycolipids, than the vehicle control. We also found that CD1d-blocking antibody treatment did not affect the levels of lymphoproliferation.

### 2.5. Interleukin (IL)-10 and IL-17A in MNC Cultures

Since we detected substantial lymphoproliferative responses to sulfatide using MNCs from PLP_139–151_-induced EAE mice, we next conducted cytokine ELISAs, using MNC culture supernatants in the presence or absence of sulfatide. The amounts of interleukin (IL)-10 were decreased with an addition of sulfatide, although it did not reach a statistical difference (Figure 7). In contrast, the sulfatide incubation did not alter the levels of IL-17A production. On the other hand, neither IL-4 nor interferon (IFN)-γ production was seen in the culture supernatants.

## 3. Discussion

### 3.1. Antibodies against GM1, GM3, GM4 and Sulfatide in RR-EAE and Human Diseases

In this study, we examined the induction of anti-glycolipid antibody responses, using serum samples from five EAE models with distinct disease courses and a viral model of MS, TMEV-IDD. Among the models, only PLP_139–151_-induced EAE mice with an RR disease course had significantly increased levels of anti-GM1, anti-GM3, and anti-GM4 antibodies; anti-sulfatide antibodies were detected in all control and MS model mice. In humans, although anti-glycolipid antibodies have been reported in various neurological diseases, it is largely unknown whether the antibodies were produced secondary to nerve damage or play a role in disease progression or recovery from the diseases [25]. In the current MS model study, it was also unclear whether the anti-glycolipid antibodies could be the cause of an RR-disease course or the result of CNS pathophysiology, although anti-glycolipid antibodies could be biomarkers for RR-EAE.

In human neurological diseases, the presence of anti-GM1 antibody has been demonstrated in acute motor axonal neuropathy (AMAN) with an antecedent infection of *Campylobacter jejuni*. Molecular mimicry between lipooligosaccharides on the surface of *C. jejuni* and GM1 on neural cells has been reported to induce cross-reactive antibody responses [28]. Although anti-GM3 and anti-GM4 antibodies have been associated with several diseases, including GBS, PP-MS, narcolepsy, type 1 diabetes mellitus, and breast cancer [29,30], neither anti-GM3 nor anti-GM4 antibody has been used as a diagnostic marker or thought to be etiologically valuable.

Anti-sulfatide antibody responses have been shown in GBS and chronic inflammatory demyelinating polyneuropathy (CIDP) [31,32], although characteristic clinical features of and diagnostic values for patients with serum anti-sulfatide antibody were inconclusive. Giannotta et al. [33] found increased anti-sulfatide IgM antibodies in different peripheral neuropathy patients, where they were often associated with a concomitant reactivity to myelin-associated glycoprotein (MAG); a selective reactivity to sulfatide was rarely found, and an association with different forms of neuropathy limited its usefulness in the diagnosis of neuropathy. Rinaldi et al. [34] reported increases in anti-sulfatide-complex antibodies (e.g., antibody against asialo-GM1:sulfatide complex), but not in antibodies against a single sulfatide. In MS, Kanter et al. [35] found that anti-sulfatide antibodies were increased in cerebrospinal fluid and that administration of sulfatide-specific antibody O4 exacerbated EAE, suggesting a pathogenic role of anti-sulfatide antibody [36]. On the other hand, administration of the same sulfatide-specific antibody O4 in the TMEV model promoted remyelination [37]; O4 is a natural IgM antibody (see Section 3.4 for more discussion on natural antibody O4). The effect of sulfatide immunization in EAE was also inconsistent; Kanter et al. [35] found exacerbation of PLP_139–151_-induced EAE, and Jahng et al. [38] found prevention of MOG_35–55_-induced EAE by sulfatide immunization.

### 3.2. Implications of Anti-Glycolipid Antibodies for Remissions/Relapses in RR-EAE

Anti-glycolipid antibodies seemed unrelated to a progressive disease course (or lack of remission) because of the absence of glycolipid antibodies in A.SW mice with PP-EAE or C57BL/6 mice with chronic EAE. In independent experiments, we found that PLP_139–151_ sensitization alone was insufficient to induce the glycolipid antibodies. As shown in Table 1, we induced hyperacute EAE in SJL/J mice by co-injection with PLP_139–151_ and curdlan; the mice developed acute fatal EAE (EAE score, 5) with no remission [18]. In hyperacute EAE mice, antibodies against GM1, GM3, and GM4 were below the detection limit, and anti-sulfatide antibody levels were lowest among MS models. Anti-glycolipid antibody levels (Abs_492nm_ ± SEM, *n* = 5) were as follows: anti-GM1, 0.019 ± 0.014; anti-GM2, 0.05 ± 0.034; anti-GM4, 0.036 ± 0.021; and anti-sulfatide, 0.363 ± 0.05. On the other hand, we found comparable levels of anti-PLP antibodies in hyperacute EAE and RR-EAE without a statistical difference (Abs_492nm_ ± SEM: hyperacute EAE, 3.36 ± 0.31; and RR-EAE, 3.70 ± 0.07). Thus, hyperacute EAE mice had sufficient time to induce antibody responses; early fatality cannot explain the lack of detectable anti-glycolipid antibodies in hyperacute EAE.

In RR-EAE, we also detected the antibody responses to the three glycolipids (GM1, GM3, and GM4) during the early stage of EAE, 14–21 days p.i. (Figure 5). Since EAE mice with high disease severity had low or undetectable anti-glycolipid antibody levels, we examined whether the anti-glycolipid antibody levels (Abs_492nm_) were correlated with EAE scores, using the Spearman’s rank correlation [39,40] (Appendix A). The correlations did not reach statistical significances, when we analyzed all sample data. However, when the data of one mouse who did not develop EAE were excluded, anti-GM1 antibody levels had a moderate negative correlation with EAE scores (*r_s_* = −0.51, *p* < 0.05); when we examined the data from EAE mice with hind limb paralysis, anti-GM1 (*r_s_* = −0.74, *p* < 0.01) and anti-GM4 (*r_s_* = −0.6, *p* < 0.05) antibody levels had negative correlations with EAE scores. These results could imply that these anti-glycolipid antibodies did not play a pathogenic role in the early stage of RR-EAE. On the other hand, since anti-glycolipid antibodies, including anti-sulfatide antibody, were also detectable during the late stage of EAE, they may function as autoreactive antibodies damaging the CNS, as suggested by Kanter et al. [35].

The time course study of serum anti-glycolipid antibody levels in individual mice may also help gain insights into the role of antibodies in relapses and remissions. However, this approach is insufficient to determine their causal or temporal relationship because of the following limitations. (1) As a principle, any descriptive correlation analyses cannot determine a causal relationship, regardless of the timing of sample collections [41]. Functional roles of these anti-glycolipid antibodies can be determined by mechanistic experiments, such as the establishment of hybridomas producing monoclonal antibodies (mAbs) of these antibodies, following the transfer of mAbs to experimental mice, as we reported previously [27]. (2) In human neurological diseases, including GBS, the positive rate of anti-glycolipid antibodies, rather than the antibody levels, have been associated with clinical signs/symptoms [42]; it is inconclusive whether serum anti-glycolipid antibody levels could be associated with severities or the time course of diseases [43,44,45,46].

### 3.3. Detrimental Roles of Anti-Glycolipid mAbs in TMEV-IDD and MOG_92–106_-Induced EAE

Although autoantibodies can be induced in various mechanisms, two mechanisms have been proved experimentally: molecular mimicry and natural antibody [37]. Molecular mimicry between host antigens and microbes has been shown to induce cross-reactive immune responses, which were initially generated against microbes that also reacted with neural components, leading to neural damage. Fujinami et al. [47] raised a mAb H8 that reacted with both TMEV and GC from spleen cells of TMEV-infected BALB/c mice. The mAb H8 reacted with the viral capsid protein VP1, neutralizing TMEV in vitro. H8 also reacted with GC, capric, lauric, and other parts of the fatty acid chain, but not galactose; when H8 was injected into mice with EAE, exacerbating demyelination. Using the Ig fraction of sera from TMEV-IDD, Fujinami’s group found that anti-TMEV antibody cross-reacted with GC, suggesting antibody(s) of the H8 type was generated and could contribute to demyelination in vivo in TMEV infection [48]. Although we observed higher anti-GC antibody levels in TMEV-IDD (Figure 3A), the titer did not reach a statistical difference, compared with the uninfected control group. The discrepancy between our current results and Fujinami’s results could be due to methodological differences, such as sample preparation (i.e., whole sera versus the Ig fraction).

Previously, we established hybridomas from A.SW mice with MOG_92–106_-induced progressive EAE [27]; in MOG_92–106_-induced PP-EAE, we observed substantial antibody deposition in the CNS demyelinating lesions, suggesting the pathogenic role of MOG_92–106_-specific antibody [20]. MOG_92–106_-specific mAbs have been shown to react with other antigens, including kidney proteins and glycolipids: GM1, GM3, and GD1b (polyreactivity). The MOG_92–106_-specific mAbs were categorized as natural antibodies [49], since (1) the subclass was IgM; (2) mAbs were reactive with a variety of self- and nonself-antigens (polyreactivity); and (3) the variable regions were encoded by germline *Ig* genes. Injection of the MOG_92–106_-specific mAb-producing hybridomas into naïve mice resulted in renal pathology, suggesting that MOG_92–106_-specific natural mAbs could play pathogenic roles in the CNS and kidney [50]. On the other hand, in our current study, the levels of anti-glycolipid antibodies were low in MOG_92–106_-induced EAE (Figure 3C). This could be due to the deposition of MOG_92–106_-specific antibodies to the CNS and other organs as demonstrated previously [20,27] or the lower concentrations of MOG_92–106_-specific IgM antibody in diluted serum samples at 1:64, compared with that of MOG_92–106_-specific IgM mAb-producing hybridoma supernatants used in the previous studies [27].

### 3.4. Potential Beneficial Roles of Natural mAbs in MS and MS Models

PLP_139–151_-induced EAE mice had humoral responses to the GM1, GM3, and GM4; the EAE mice did not mount cellular responses to the three glycolipids. These results suggested that anti-GM1, anti-GM3, and anti-GM4 antibodies could be natural antibodies produced in a T-cell independent manner. Although we were not able to determine the subclass of these antibodies, the subclass of anti-sulfatide antibody was IgM (Figure 4B). On the other hand, we found lymphoproliferative responses to sulfatide with a decrease in IL-10 production; anti-sulfatide lymphoproliferative responses were independent of CD1d molecules, although CD1d molecules have been shown to present glycolipid antigens, including sulfatide, to natural killer T (NKT) cells [26]. Our results were consistent with the findings by Matsumoto et al. [51], who demonstrated anti-glycolipid antibody induction in CD1d deficient mice without glycolipid-specific NKT cell stimulation; their results showed that anti-glycolipid antibody production was independent of CD1d or NKT cells. Although the sensitivity of glycolipid-specific lymphoproliferative assays could be improved by enrichment of NKT cells with CD1d-positive antigen-presenting cells, this is not necessarily the case of glycolipid antigens in EAE, since (1) glycolipid-specific T cells in the CNS of EAE mice have been reported to be distinct from invariant NKT cells [38]; (2) CD1d-restricted cells include not only NKT cells, but also αβ T cells, and γδ T cells [52]; (3) glycolipids, including sulfatide, can affect T cell proliferation independent of CD1d presentation in EAE [53]; and (4) glycolipid-specific αβ T cells isolated from MS patients were restricted to CD1b, which were not encoded in mice [54].

To determine whether these antibodies are natural antibodies, one needs to test their polyreactivity, determine Ig subclass, and sequence the variable regions of these antibodies, which have been examined following the establishment of hybridomas producing natural mAbs. In the above Section 3.3, we discussed the two natural mAbs established previously from MOG_92–106_-induced EAE and TMEV-IDD, in which the natural mAbs seemed to play a pathogenic role. In contrast, several IgM mAbs (SCH94.03, SCH79.08, O1, O4, and HNK-1) have also been shown to have characteristics of natural antibodies [37,55,56,57]; some of these mAbs, including sulfatide-specific O4, seemed to play a protective/beneficial role, promoting remyelination, and reducing the relapse rate and demyelination in EAE mice [58]. Clinically, Kirschning et al. [59] established a clone of IgM mAb, DS1F8, with characteristics of natural antibody from an MS patient. Matsiota et al. [60] found that MS patients often had elevated antibody levels against many autoantigens in cerebrospinal fluid, suggesting the production of natural autoantibodies. Although it is unknown whether these antibodies have any effects on demyelination, it is attractive to hypothesize that the presence or absence of remyelination-promoting natural antibodies in MS or its models might alter the disease courses [37].

### 3.5. Roles of Epitope Spreading for Induction of Anti-Glycolipids Antibodies

Another mechanism by which anti-glycolipid antibodies could be produced is epitope (or determinant) spreading. Epitope spreading is the development of immune responses to endogenous epitopes secondary to the release of self-antigens during a chronic autoimmune or inflammatory response [61]. Epitope spreading has been suggested to play a pathogenic role only during the late chronic stage of MS models following the destruction of the myelin sheaths. Theoretically, in PLP_139–151_-induced EAE, PLP_139–151_-specific T cells initially invade the CNS and damage PLP-positive CNS tissues, releasing CNS glycolipid antigens into the periphery. Then, antigen-presenting cells present the released-glycolipid antigens (for example, via CD1d molecules), which can be followed by the induction of anti-glycolipid immune responses. Autoantibodies generated by epitope spreading might be involved in disease relapses. In the current experiment, since we detected anti-GM1, anti-GM3, anti-GM4, and anti-sulfatide antibodies as early as 14 days p.i.; thus, these antibodies were likely produced independent of epitope spreading, although epitope spreading might contribute to anti-glycolipid antibody induction during the late stage of MS models.

## 4. Materials and Methods

### 4.1. Mice

We purchased 4-week-old female SJL/J mice from the Jackson Laboratory Japan, Inc. (Yokohama, Japan), C57BL/6J mice from CLEA Japan, Inc. (Tokyo, Japan), and A.SW mice from the Jackson Laboratory (Bar Harbor, ME, USA). Mice were maintained under a specific-pathogen free environment in the animal breeding facility of Kindai University Faculty of Medicine, Osaka, Japan. All experimental procedures were approved by the Institutional Animal Care and Use Committee of Kindai University Faculty of Medicine (KAME-2021-006) and performed according to the criteria outlined by the National Institutes of Health (NIH) National Research Council [62].

### 4.2. Induction and Evaluation of EAE

We induced EAE by subcutaneous (s.c.) sensitization with 100 nmol/mouse of modified PLP_139–151_ peptide (VSLGKWLGHPDKF, United BioSystems, Herndon, VA, USA) in SJL/J mice, MOG_92–106_ peptide (DEGGYTCFFRDHSYQ, United BioSystems) in SJL/J mice or A.SW mice, or MOG_35–55_ (MEVGWYRSPFSRVVHLYRNGK, United BioSystems) peptide in C57BL/6 mice, in which myelin peptides were emulsified in CFA composed of incomplete Freund’s adjuvant (BD, Franklin Lakes, NJ, USA] and *Mycobacterium tuberculosis* H37 Ra (BD). The final concentration of *M. tuberculosis* in the myelin peptide/CFA emulsion was 2 mg/mL (400 μg/mouse). In MOG_92–106_-sensitized SJL/J mice, we injected intraperitoneally (i.p.) with 5 mg of curdlan (FUJIFILM Wako Pure Chemical Corporation, Osaka, Japan) in 200 μL of phosphate-buffered saline (PBS) one day before MOG sensitization [18]. To induce chronic MOG_35–55_-induced EAE, we injected C57BL/6 mice twice with MOG_35–55_ s.c. on days 0 and 19 and 300 ng of pertussis toxin (List Biological Laboratories, Campbell, CA, USA) i.p. on days 0 and 2 [63]. Classical EAE signs were assessed as follows: 0, no sign; 1, tail paralysis; 2, mild hindlimb paresis; 3, moderate hindlimb paralysis; 4, complete hindlimb paraplegia; and 5, quadriplegia, moribund state, or death. Ataxic EAE signs observed in MOG_92–106_-sensitized mice were assessed as follows: 0, no sign, 1 or 2, mice turning their heads or bodies to one side, scored as 1 or 2 depending on the degree to which the head was turned; 3, mice continuously rolled by twisting their bodies or rotated laterally in a circle; 4, mice could not stand but would lay on their sides; and 5, moribund state or death [20].

### 4.3. Induction and Evaluation of TMEV-IDD

We induced a viral model of MS by intracerebral injection of TMEV. SJL/J mice were inoculated with 2 × 10^5^ plaque-forming units (PFUs)/mouse of the Daniels (DA) strain of TMEV. Clinical scoring of TMEV-IDD was based on impairment of the righting reflex as follows: 0, the mouse resists being turned over; 1, the mouse is flipped onto its back but immediately rights itself on one side; 1.5, the mouse is flipped onto its back but immediately rights itself on both sides; 2, the mouse rights itself in 1 to 5 s; 3, righting takes more than 5 s; and 4, the mouse cannot right itself [18].

### 4.4. ELISAs for Antibodies against Glycolipids

Unless otherwise noted, we harvested sera from MS model mice when we completed the observation of their clinical courses: PLP_139–151_-induced EAE, 70 days p.i.; MOG_35–55_-induced EAE, 1 month p.i.; MOG_92–106_-induced EAE in A.SW or SJL/J mice, 1–3 months p.i.; TMEV-IDD, 1–2 months post infection. Using serum samples, we conducted ELISAs to detect antibodies against 11 glycolipids: GM1, GM2, GM3, GM4, GD3, GC, GD1a, GD1b, GT1b, GQ1b, and sulfatide (Sigma-Aldrich, St. Louis, MO, USA), as described previously. Glycolipid antigens were dissolved in methanol/chloroform (1:1) and diluted with 100% ethanol. Flat-bottom Corning^®^ 96-well ELISA Microplates (Corning Incorporated, Corning, NY, USA) were coated with the glycolipid antigen solutions at a concentration of 200 ng/well. After blocking with 1% bovine serum albumin (BSA) in PBS, serum samples diluted with the blocking solution at a ratio of 1:64 were added to the wells. The microplates were incubated for 90 minutes (min) at room temperature (RT) and then washed with 0.1% BSA in PBS. We added peroxidase-conjugated anti-mouse F(ab′)_2_ antibody (5000-fold dilution, Jackson ImmunoResearch Laboratories, Inc., West Grove, PA, USA) to the wells and incubated for 90 min at RT. After washing the plates with 0.1% BSA in PBS, a substrate solution composed of 0.4 mg/mL of *o*-phenylenediamine dihydrochloride and 0.006% H_2_O_2_ in PBS was added to the wells. Following a 15-min incubation at RT, the reaction was stopped by adding 8N H_2_SO_4_. Using the BioTek Synergy H1 Hybrid Multi-Mode Microplate Reader (Agilent Technologies, Inc., Santa Clara, CA, USA), we measured anti-glycolipid antibody levels at 492 nm (Abs_492nm_) by subtracting the absorbances of glycolipid-uncoated wells from those of glycolipid-coated wells. To determine the isotype of anti-glycolipid antibodies, we conducted anti-glycolipid ELISAs, using peroxidase-conjugated IgM-specific (2500-fold dilution, Enzo Life Sciences Inc., Farmingdale, NY, USA), or IgG-specific antibody (500-fold dilution, Thermo Fisher Scientific Inc., Waltham, MA, USA).

### 4.5. Lymphoproliferative Responses and Cytokine ELISAs

We harvested the spleen and inguinal lymph nodes from PLP_139–151_-EAE mice, mashed the organs on metal mesh with 50-μm pores using a plunger of a 5-mL syringe to make single-cell suspensions, and isolated MNCs from the single-cell suspensions using Histopaque^®^-1083 (Sigma-Aldrich, Co., St. Louis, MO, USA). MNCs were cultured at 2 × 10^5^ cells/well in 96-well flat-bottom plates in a triplicated manner and stimulated with 5 μg/mL of glycolipid antigens (GM1, GM3, GM4, GD1b, and sulfatide) or the vehicle control (methanol:chloroform = 1:1) in the presence or absence of 10 μg/mL of anti-CD1d mAb (Thermo Fisher Scientific Inc.) at 37 °C with 5% CO_2_ for 5 days. To determine the glycolipid-specific lymphoproliferative responses, we added 3 μL of a CCK-8 solution (Dojindo Laboratories, Kumamoto, Japan) to the wells of the plate for the last 24 hours, as reported previously [18]. We measured the absorbances at 450 nm (Abs_450 nm_) of triplicate wells, using the Synergy H1 Hybrid Multi-Mode Microplate Reader. The Abs_450 nm_ of lymphoproliferation without stimulation (autoproliferation) ranged from 0.12 to 0.29, regardless of mouse strains, different autoantigen sensitizations, TMEV infection, or no treatment.

For cytokine ELISAs, we incubated the MNCs from the spleen/inguinal lymph nodes with 5 μg/mL of sulfatide or the vehicle (methanol:chloroform = 1:1) in the presence or absence of 10 μg/mL of anti-CD1d mAb at 37 °C with 5% CO_2_ for 2 days. The culture supernatants were collected and stored at −80 °C until examined. The amounts of IL-4 (BD Biosciences, San Jose, CA, USA), IL-10 (BD Biosciences), IL-17A (Biolegend, Inc., San Diego, CA, USA), and IFN-γ (BD Biosciences) in the culture supernatants were quantified in duplicate wells, using ELISA kits according to the manufacturers’ instructions [18].

### 4.6. Statistical Analyses

We used OriginPro 2023 (OriginLab, Corporation, Northampton, MA, USA) for statistical analyses. We used Student’s *t*-test for two groups and analysis of variance (ANOVA) with post hoc Fisher’s LSD test for three or more than three groups. We correlated each antibody level with EAE scores, using the Spearman’s rank correlation [40,64]. We used the interpretation for the Spearman’s correlation by Mukaka (2012) [65] as follows: 0.7 to 1.0 (−0.7 to −1.0), high positive (negative) correlation; 0.5 to 0.7 (−0.5 to −0.7), moderate positive (negative) correlation; 0.3 to 0.5 (−0.3 to −0.5), low positive (negative) correlation; and 0 to 0.3 (0 to −0.3), negligible correlation [66]. We conducted a power analysis, using an R version 4.3.0 and the package “WebPower” version 0.9.3 [67,68].

## 5. Conclusions

Among five autoimmune and one viral models of MS, we detected anti-glycolipid antibodies only in RR-EAE. Although it is unknown whether anti-glycolipid antibodies could be associated with relapses or remissions of the disease, anti-glycolipid antibodies may be useful as a biomarker for RR-MS.

## Figures and Tables

**Figure 1 ijms-24-12937-f001:**
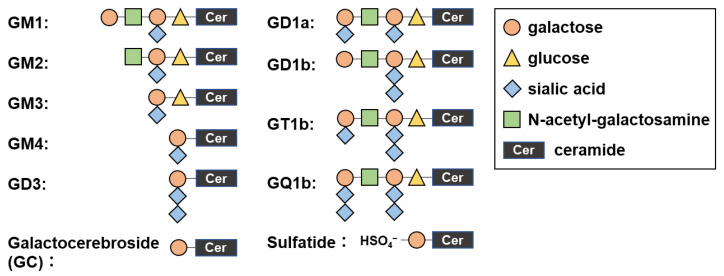
Structures of representative glycolipids. The saccharide chains are composed of galactose, glucose, sialic acid, or N-acetyl-galactosamine, which bind ceramide. Ceramides are lipids composed of amino alcohol and fatty acid varying in length. Shown are 11 glycolipids that we used in the current study as antigens and determined anti-glycolipid antibody levels by enzyme-linked immunosorbent assays (ELISAs).

**Figure 2 ijms-24-12937-f002:**
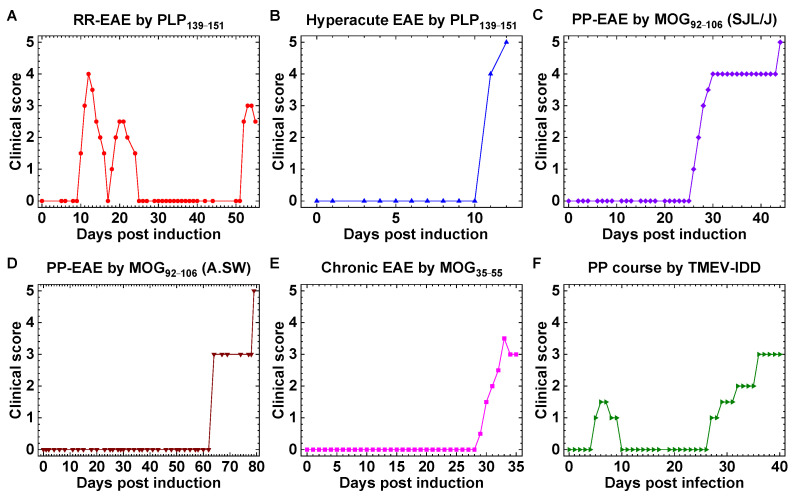
Multiple sclerosis (MS) models with distinct clinical courses. (**A**) Myelin proteolipid protein (PLP)_139–151_-sensitized SJL/J mice developed relapsing-remitting (RR) experimental autoimmune encephalomyelitis (EAE). (**B**) PLP_139–151_-sensitized SJL/J mice with curdlan injection developed hyperacute EAE, and all mice died around 2 weeks post-induction (p.i.). (**C**) Myelin oligodendrocyte glycoprotein (MOG)_92–106_-sensitized SJL/J mice with curdlan injection developed primary progressive (PP) EAE without remission. (**D**) MOG_92–106_-sensitized A.SW mice developed chronic PP-EAE. (**E**) MOG_35–55_-sensitized C57BL/6 mice developed chronic EAE 1 month p.i. (**F**) Theiler’s murine encephalomyelitis virus (TMEV)-infected SJL/J mice developed a PP disease, TMEV-induced demyelinating disease (TMEV-IDD), around 1 month after virus inoculation (chronic phase). Although TMEV-infected mice also had neurological signs 1 week after infection (acute phase), the acute disease was induced by direct virus infection in the gray matter, leading to polioencephalomyelitis, not by demyelination in the white matter. Shown are representative disease courses.

**Figure 3 ijms-24-12937-f003:**
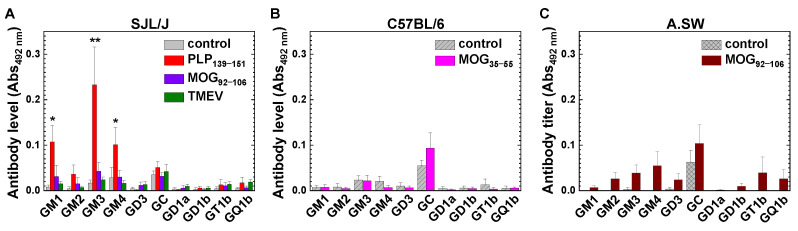
Anti-glycolipid antibody levels in MS models and untreated naïve mice (control). Using anti-mouse F(ab’)_2_ antibody that detects all immunoglobulin (Ig) subclasses, we conducted ELISAs to examine the levels of antibodies against 10 glycolipids: GM1, GM2, GM3, GM4, GD3, galactocerebroside (GC), GD1a, GD1b, GT1b, and GQ1b. We measured the absorbances at 492 nm (Abs_492nm_) in glycolipid-coated and uncoated wells, and subtracted the absorbances of uncoated wells from those of glycolipid-coated wells. The data were the mean Abs_492 nm_ + standard error of the mean (SEM) of serum samples. (**A**) Anti-glycolipid antibody levels of SJL/J mice. We detected higher anti-GM1, anti-GM3, and anti-GM4 antibodies in PLP_139–151_-sensitized EAE mice than the control mice (*, *p* < 0.05; and **, *p* < 0.01) (control, *n* = 8; PLP_139–151_, *n* = 8; MOG_92–106_, *n* = 6; and TMEV-IDD, *n* = 13). (**B**) Anti-glycolipid antibody levels of C57BL/6 mice. We did not detect significantly high anti-glycolipid antibodies compared with the control mice (control, *n* = 8; and MOG_35–55_, *n* = 8). (**C**) Anti-glycolipid antibody levels of A.SW mice. We did not detect significantly high anti-glycolipid antibodies compared with the control mice (control, *n* = 3; and MOG_92–106_, *n* = 8).

**Figure 4 ijms-24-12937-f004:**
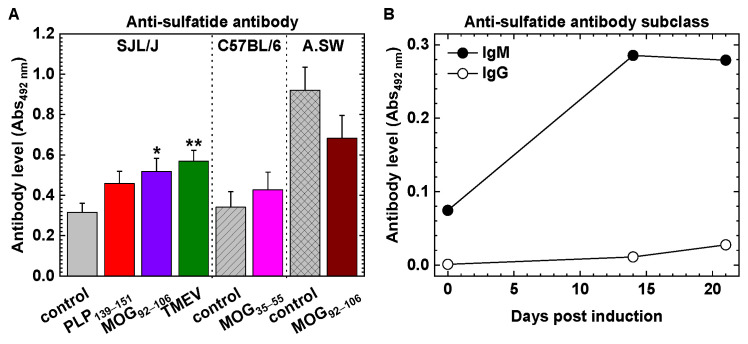
Anti-sulfatide antibody and its subclass. (**A**) Anti-sulfatide antibodies were detected in all controls and MS models. SJL/J mice: control, *n* = 8; PLP_139–151_, *n* = 8; MOG_92–106_, *n* = 6; and TMEV-IDD, *n* = 13 (*, *p* < 0.05; and **, *p* < 0.01). C57BL/6 mice: control, *n* = 8; and MOG_35–55_, *n* = 8. A.SW mice: control, *n* = 3; and MOG_92–106_, *n* = 8. The data were the mean Abs_492nm_ + SEM of serum samples. (**B**) Using sera from PLP_139–151_-sensitized SJL/J mice, we conducted ELISAs with anti-mouse IgM (●) or IgG (○) Fc-specific antibody as a detection antibody. Anti-sulfatide antibody was elevated at 14 days p.i. of EAE, and its subclass was IgM.

**Figure 5 ijms-24-12937-f005:**
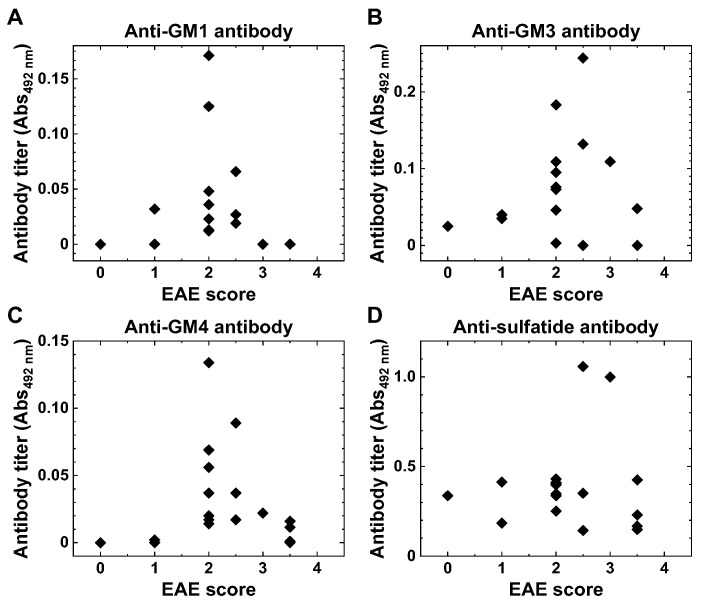
Anti-glycolipid antibody levels and EAE severities. We collected serum samples on 14–21 days p.i. from PLP_139–151_-induced RR-EAE mice (*n* = 18) and evaluated the associations between anti-glycolipid antibody levels and EAE scores. Anti-GM1 (**A**), anti-GM3 (**B**), and anti-GM4 (**C**) antibody levels were undetectable or low in EAE mice with high disease severity (EAE score 3 or more). (**D**) Anti-sulfatide antibody levels were higher than the other anti-glycolipid antibodies, regardless of disease severity.

**Figure 6 ijms-24-12937-f006:**
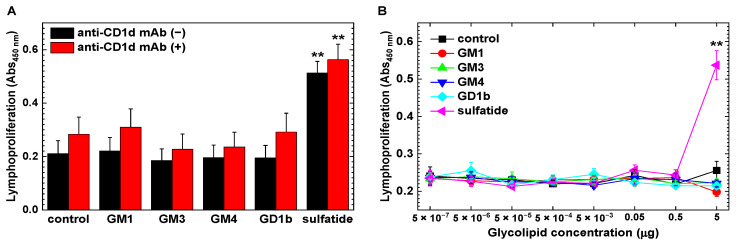
Lymphoproliferative assays against glycolipids. We harvested mononuclear cells (MNCs) from PLP_139–151_-induced EAE mice. Using the Cell Counting Kit-8 (CCK-8), we examined lymphoproliferative responses in the presence or absence of glycolipids: GM1, GM3, GM4, GD1b, or sulfatide. To assess whether glycolipid antigens were presented on CD1d molecules, we conducted the assays in the presence (+) or absence (−) of anti-CD1d monoclonal antibody (mAb). (**A**) We observed higher lymphoproliferative responses to sulfatide, but not to the other glycolipids, compared with the vehicle control (**, *p* < 0.01). Anti-CD1d mAb treatment did not affect the lymphoproliferative responses. The data were the mean + SEM of four pools of MNCs isolated from spleens and inguinal lymph nodes; one pool was from two to three mice. (**B**) We incubated MNCs with different doses of glycolipids and found significant lymphoproliferation only with sulfatide at 5 μg/mL, compared with the vehicle control (**, *p* < 0.01). MNCs were isolated 3 weeks p.i. The data were the mean Abs_450 nm_ ± SEM of two pools of MNCs. One pool was from spleens of two to three mice. We conducted the assays, using triplicate wells, and determined the statistical differences by analysis of variance (ANOVA).

**Figure 7 ijms-24-12937-f007:**
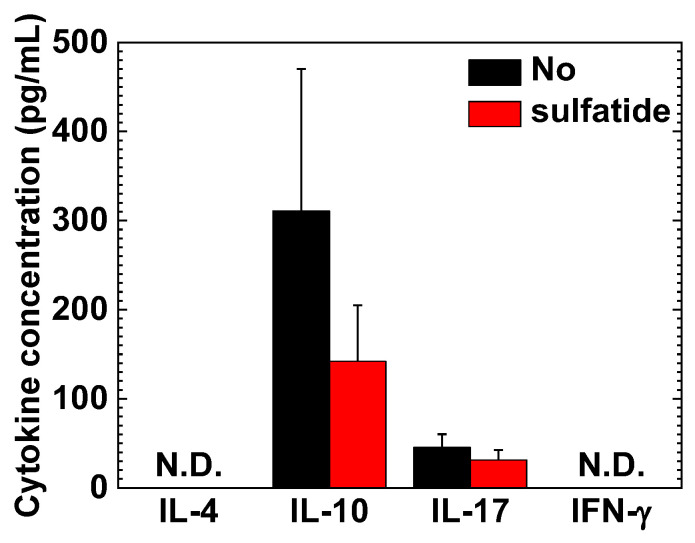
Cytokine production from sulfatide-stimulated MNCs in PLP_139–151_-induced EAE. We isolated MNCs from PLP_139–151_-induced EAE mice and incubated MNCs in the presence or absence of sulfatide. We quantified the amounts of interleukin (IL)-4, IL-10, IL-17, and interferon (IFN)-γ in the supernatants. We found that IL-10 production was suppressed by an addition of sulfatide, although it did not reach a statistical difference. IL-4 and IFN-γ were not detectable (N.D.). The data were the mean + SEM of four pools of spleen and inguinal lymph nodes from two to three mice.

**Table 1 ijms-24-12937-t001:** Animal models of multiple sclerosis with distinct clinical courses.

Clinical Course	Inoculum	Adjuvant	Mouse Strain	Model	Reference
RR	PLP_139–151_	CFA	SJL/J	EAE	[18]
hyperacute	PLP_139–151_	CFA/curdlan	SJL/J	EAE	[18]
PP, SP	MOG_92–106_	CFA/curdlan	SJL/J	EAE	[19]
PP	MOG_92–106_	CFA	A.SW	EAE	[20]
monophasic/chronic	MOG_35–55_	CFA, PT	C57BL/6	EAE	[23]
PP	TMEV	–	SJL/J	virus	[18]

Abbreviations: CFA, complete Freund’s adjuvant; EAE, experimental autoimmune encephalomyelitis; MOG, myelin oligodendrocyte glycoprotein; PLP, myelin proteolipid protein; PP, primary progressive; PT, pertussis toxin; RR, relapsing-remitting; SP, secondary progressive; and TMEV, Theiler’s murine encephalomyelitis virus.

## Data Availability

Not applicable.

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
