# Peer review of "Anti-Glycolipid Antibody Examination in Five EAE Models and Theiler’s Virus Model of Multiple Sclerosis: Detection of Anti-GM1, GM3, GM4, and Sulfatide Antibodies in Relapsing-Remitting EAE"

_ijms, 2023, doi:10.3390/ijms241612937_

Round 1
Reviewer 1 Report (Previous Reviewer 1)
Upon reviewing the changes that have been made, I think that the manuscript has been enhanced.

Author Response
We appreciate the reviewer who has accepted our manuscript as the current form.
Reviewer 2 Report (New Reviewer)
The authors wrote a comprehensive article about Anti-Glycolipid Antibodies in Five EAE Models and Theiler’s Virus Model of Multiple Sclerosis.
However the only thing that boders me is the title that is somewhat confusing. I first have to look at the abstract to understand what the authors ment by the title. So I think this could be an issue.
THE TITLE: Anti-Glycolipid Antibodies in Five EAE Models and Theiler’s Virus Model of Multiple Sclerosis; Anti-GM1, GM3, GM4, and Sulfatide Antibody Detection in Relapsing-Remitting EAE
Maybe the authors should consider a different title without the any detection, so it could be something like this:
Anti-Glycolipid Antibodies examination in Five different EAE Models and Theiler’s Virus Model of Multiple Sclerosis: detection of Anti-GM1, GM3, GM4, and 3 Sulfatide Antibody Detection in Relapsing-Remitting EAE
Other part of the article are very good organized: introduction, results, methods, discussion, conclusion.
Author Response
We appreciate the comment. Accordingly, we have changed the title as follows:
Old
Anti-Glycolipid Antibodies in Five EAE Models and Theiler’s Virus Model of Multiple Sclerosis; Anti-GM1, GM3, GM4, and Sulfatide Antibody Detection in Relapsing-Remitting EAE
New
Anti-Glycolipid Antibody Examination in Five EAE Models and Theiler’s Virus Model of Multiple Sclerosis; Detection of Anti-GM1, GM3, GM4, and Sulfatide Antibodies in Relapsing-Remitting EAE
This manuscript is a resubmission of an earlier submission. The following is a list of the peer review reports and author responses from that submission.
Round 1
Reviewer 1 Report
The current paper entitled "Anti-Glycolipid Antibodies in Five EAE Models and Theiler's Virus Model of Multiple Sclerosis; Possible Roles of the Antibodies for Relapses and Remissions" aimed to investigate the link between multiple sclerosis (MS) and anti-glycolipid antibodies and determine whether these antibodies are related to the disease etiology, prognosis, relapsing or remission.
In the introduction, Komlós et al. reviewed previous studies in reasonable depth and used relevant references with limited self-citation. The research gaps were mentioned in the introduction clearly.
Due to the heterogeneity in multiple sclerosis, Moriguchi et al. employed various animal MS models with different disease courses (relapsing-remitting, primary progressive, secondary progressive, and monophasic) and etiologies (experimental autoimmune encephalomyelitis (EAE) versus Theiler's murine encephalomyelitis virus (TMEV)) to reach more conclusive results and fulfill the study's aim. Then, they quantified the serum levels of 11 anti-glycolipid antibodies.
The only model that showed a significant increase in four anti-glycolipid antibodies is the relapsing-remitting EAE induced by myelin proteolipid protein 139-151 (PLP139-151). The four anti-glycolipid antibodies are GM1, GM3, GM4, and sulfatide. Furthermore, in isolated mononuclear cells from PLP139-151-EAE mice, sulfatide, but not other glycolipids, induced CD1d-independent lymphoproliferative responses with decreased interleukin (IL)-10 production. The authors also used some experimental controls for proper comparison. The results were presented very well with appropriate figure captions. The discussion and conclusion were parallel with the results. Overall, this manuscript showed original work and is worth publication in its current version.
Author Response
We appreciate the reviewer who has accepted our original manuscript as the current form.
Reviewer 2 Report
Here Nakamura K et al. analyzed the circulating levels of anti-glycoprotein antibodies in different mice models of multiple sclerosis. The authors report the presence of anti-GM1, -GM3, and GM-4 antibodies in the Relapsing Remitting-EAE course.
Strengths: Overall this is an interesting and clinically relevant topic and prior studies in human multiple sclerosis and other neuropathies have reported the anti-glycolipid antibodies in serum and CSF fluid. Here the authors try to corroborate this human clinical data with the experimental studies in relevant mice models with different clinical spectra.
Weaknesses: The study lacks analysis at different time points. All of these mice models do have different timescales for their clinical manifestations. To conclude definitively, the authors must compare anti-glycolipid antibody levels in a kinetic fashion and not a single time point, which may be an ideal time for one model but not the other. There are several conclusions that authors have made in the manuscript with no supportive data and some of the interpretation seems ambiguous and erroneous.
Below are specific points addressing them that may help improve the overall quality and readability of the manuscript.
1. Data for PP form of EAE induced in A.SW mice and TMEV-induced EAE in SJL/J mice are missing.
2. Fig 2D. The complete remission of the clinical symptoms in MOG35-55 peptide/CFA-induced EAE in C57BL/6 mice in a matter of 3 days is inconsistent with what the majority of the studies have reported. There are endless papers that conclusively demonstrated a typical ascending flaccid paralysis course with chronic/sustained neuroinflammation.
3. Fig 3. The sera samples were collected at the endpoint of the corresponding Figure 1 timeline. Since each of the model systems has a different timeline of clinical symptoms, it becomes highly imperative to compare antibody levels in a kinetic fashion and not a single time point.
4. Fig 5. The author's conclusion of lower levels of anti-glycolipid antibodies in mice with higher clinical grade scoring is ambiguous and does not support by the data presented in Fig 5. It should be noted that mice with lower scores (0 and 1) also had correspondingly lowered levels of circulating anti-GM1, anti-GM3, and anti-GM4 antibodies. Moreover, the data points at score 2 are highly scattered, indicating that some mice had higher antibodies than others despite having similar clinical symptoms. Further, this interpretation was solely based on a few data points that don't justify the statistical vigor of the data. A correlation analysis has to be performed to conclusively demonstrate the correlation between the circulating anti-glycolipid levels and the EAE clinical score.
5. Fig 6. The MNCs derived from the spleen and lymph nodes of PLP139-151-induced EAE mice seem to be proliferative anyways regardless of glycolipid or their presentation on anti-Cd1d. The authors need to include MNC from naive mice as a baseline control for the lymphoproliferative reaction. In a mixed population of MNC, it may be possible that glycolipids can modulate lymphocyte response in a direct or indirect way through other cells. Also, on what time scale the lymphoproliferation is assessed is unclear.
6. Discussion 3.2. The data presented in the manuscript, do not support in any way the author's conclusion that anti-GM1, 3, and 4 antibodies may have a protective role in the RR-EAE course in SJL/J mice.
7. The statement "Although the levels of anti-314 glycolipids antibodies were low in MOG92-106-induced EAE in our current study (Figure 315 3C), this could be due to the deposition of the antibodies to target organs or the lower 316 concentrations of IgM in serum samples, compared with purified IgM mAb used in the 317 previous studies" seems to be a speculative and not supported by the data.
8. The data corresponding to the statement "Although we observed higher anti-GC antibody levels in TMEV-IDD, the titer did not 326 reach a statistical difference, compared with the uninfected control group" is not found in the manuscript.
9. The phrase "data not shown" is not helpful in the manuscript. The authors can use a space in the supplementary figures to show all the relevant data that help improves the readability of the manuscript.
Round 2
Reviewer 2 Report
While the authors have adequately addressed some of the minor issues/concerns, the key issues remain unanswered in the revised manuscript. Overall, the authors have measured anti-Glycolipid Abs in various EAE models of varying clinical course and severity, however, the vast majority of authors' claim or conclusion is not supported by the data, and many of the experiments lack proper controls. For instance, lymphoproliferative data did not take into account the overall proliferative response in EAE mice and lacked control from a naive age-matched mouse to justify whether anti-glycolipid Abs regulate lymphocyte response.
The following key weaknesses are still not addressed.
1. The primary claim of the authors is that anti-Glycolipid Abs levels play a role in remission and relapse dynamics of the RR-EAE model, and the data presented in the revised manuscript do not support it at all. There isn't any comparison of Ab levels in various relapse and remission phases, nor there is an attempt to explain how changes in the clinical course and measured Ab titers are associated. It should be imperative to show how anti-glycolipid Ab levels change during the different clinical phases of a given EAE model, without such comparison it would be insufficient evidence to comment on the possible role of these Abs in shaping the clinical course.
2. The author's correlation analysis of anti-glycolipid Ab levels and EAE severity look confusing and ambiguous. it is completely unclear how that correlation was measured against clinical severity nor it is clear how authors have explained the similar Abs levels corresponding to the different grades of the EAE in Figure 5.
3. The author's justification about the possible role protective of anti-Glycolipid Abs in the EAE course solely based on their levels below the detection limits in the samples derived from EAE mice that received curdlan seems ambiguous and does not stand anywhere close to the rationale or logical explanation.
Round 3
Reviewer 2 Report
While the authors did a good job to address some of the critical issues in the manuscripts, their primary interpretation that anti-glycolipid antibodies might play a role in relapse or remission of the disease course in RR-EAE models solely based on their induction in those EAE mice but not in other EAE models seems elusive. It is highly likely that different peptide immunization protocols with different immunodominant myelin epitopes may invoke different pathogenic mechanisms. Further, the frequency of cognate BCR-harboring autoimmune B cells may also skew such differential patterns. More importantly, the lack of detectable levels of Abs in the PLP139-151+curdlan group might be due to hyper-inflammatory response and early fatality, which might not allow sufficient time to build and maintain circulating anti-glycolipid Ab levels.
Section 3.2. The author's interpretation that anti-glycolipid Abs were irrelevant to SJL/J mice strains because PLP139-151 but not MOG92-106-mediated immunization induce them seems confusing, and argued on the grounds of the explanation given in the above paragraph.
The statement "In RR-EAE, antibody responses to the three glycolipids (GM1, GM3, and GM4) may play roles in either relapses or remissions of RR-EAE, since these antibodies were detectable during the initial stage of EAE, 14 days p.i. (Figure 5)" again seems ambiguous and misleading. There isn't any data to support their role in any course. At one place the authors discussed that it is difficult to ascribe a specific role to these Abs in a disease course and on the other end they are interpreting their potential roles in relapse or remission, or based on descriptive correlation analysis with a limited number of samples assigning them a protective role.
The lymphoproliferation performed on total mononuclear cells from the spleen and lymph nodes to assess the possible role of cellular immunity against glycolipids also need to be revisited. Unlike conventional peptide antigens, these glycolipids are primarily presented on CD1d-expressing APCs. Moreover, CD1d-restricted T cells and NKT populations are extremely rare, so analyzing their response in total MNC cell populations may not be useful to gauge the reactivity of those rare cells. The authors need to analyze responses using enriched rare APCs and -restricted T cells.
The authors need to do a lot of work to organize their thoughts around their data and interpretation and come up with a logical explanation rather than commenting on Ab's half-life and disease course timeline differences, which is really not helping the cause as the underlying Ab-mediated pathogenic mechanisms in neuroimmune disease pathogenesis involves self-sustained inflammation and couldn't be just explained by the half-life of generic antibodies.
In my opinion, in its current form, this manuscript couldn't convenience the possible role of anti-glycolipids Ab in relapse or remission and nor explain anything about the possible cellular response to these self-glycolipids.
